# An Unsupervised Fusion Strategy for Anomaly Detection via Chebyshev Graph Convolution and a Modified Adversarial Network

**DOI:** 10.3390/biomimetics10040245

**Published:** 2025-04-17

**Authors:** Hamideh Manafi, Farnaz Mahan, Habib Izadkhah

**Affiliations:** Department of Computer Science, University of Tabriz, Tabriz 5166616471, Iran

**Keywords:** anomaly, adversarial network, Numenta, Chebyshev convolution

## Abstract

Anomalies refer to data inconsistent with the overall trend of the dataset and may indicate an error or an unusual event. Time series prediction can detect anomalies that happen unexpectedly in critical situations during the usage of a system or a network. Detecting or predicting anomalies in the traditional way is time-consuming and error-prone. Accordingly, the automatic recognition of anomalies is applicable to reduce the cost of defects and will pave the way for companies to optimize their performance. This unsupervised technique is an efficient way of detecting abnormal samples during the fluctuations of time series. In this paper, an unsupervised deep network is proposed to predict temporal information. The correlations between the neighboring samples are acquired to construct a graph of neighboring fluctuations. The extricated features related to the temporal distribution of the time samples in the constructed graph representation are used to impose the Chebyshev graph convolution layers. The output is used to train an adversarial network for anomaly detection. A modification is performed for the generative adversarial network’s cost function to perfectly match our purpose. Thus, the proposed method is based on combining generative adversarial networks (GANs) and a Chebyshev graph, which has shown good results in various domains. Accordingly, the performance of the proposed fusion approach of a Chebyshev graph-based modified adversarial network (Cheb-MA) is evaluated on the Numenta dataset. The proposed model was evaluated based on various evaluation indices, including the average F1-score, and was able to reach a value of 82.09%, which is very promising compared to recent research.

## 1. Introduction

An anomaly is a sample or sequence of samples that exhibits behavior markedly distinct from that of most samples. Anomaly detection is a procedure used in several real-world applications and across diverse data sources. One of the types of samples in the real world is a time series. Anomalies in time-series data may provide crucial information across several domains, including banking, aerospace, information technology, security, and medicine. Nonetheless, anomaly identification in these data is very intricate and demanding, owing to the ambiguous definition of anomalies, the absence of sample labels, and the lack of temporal correlations. Inherent difficulties exist, including the tenuous border between normal and abnormal samples and the potential for noise in samples caused by sensor faults or erroneous readings that may mimic anomalies. Context-specific anomalies may manifest as varying behaviors within a time series within a singular process [1,2].

An anomaly in a biological context is a variation from the body’s or from genetics’ natural or normal condition. Congenital deformities, traumas, or particular physiological traits may be regarded as anomalies. Bionics, which integrates biological understanding with technology, is often used to mitigate these aberrations. Examples include bionic prosthetics, neural implants, and similar technologies. Bionics are used not just to rectify deficiencies but also to augment human capacities beyond the normative range. These may be classified as “functional anomalies,” namely brain–machine interfaces facilitating device control via cognitive processes. Anomalies in biology or technology may provide challenges or serve as inspiration for bionics, which seeks to restore, enhance, or redefine the natural condition [2,3]. The combination of bionics and anomalies can create interesting and new concepts in various sciences, especially in fields such as medical engineering, artificial intelligence, and biology. This combination suggests a future where bionic systems can intelligently adapt to different conditions and identify and correct unexpected problems [3].

Data mining effectively recognizes ambiguous patterns and is a crucial mechanism for transforming data into knowledge. A primary rationale for using data mining is effectively examining a diverse array of observations based on their behaviors [3,4]. The potential for automated sample analysis is particularly significant in the contemporary landscape, characterized by an excessive rise in both the number and variety of samples. Instances of automated analysis encompass information extraction, used in anomaly detection. When a pattern emerges among samples that deviate from typical behaviors or the predominant samples, it may be classified as an anomaly [5]. These irregular patterns are called anomalies, outliers, or inconsistent findings. Anomaly detection was first established in the statistical community in the early 19th century, and subsequently, several techniques for its identification were developed. In those days, anomaly detection was conducted entirely manually and visually by specialists and experts in each field. However, manual detection was accompanied by problems. Among the problems of manual detection to be considered are human error, high detection time, uncertainty, etc. Based on the above, automatic anomaly detection has been developed using machine learning techniques recently. Anomaly detection is now a key challenge in machine learning and deep learning. Automated anomaly detection is crucial in the contemporary landscape, where the vast quantity of samples renders human labeling unfeasible. Anomaly detection encompasses several applications, including credit card fraud, cybersecurity intrusion, defect detection in critical safety systems, insurance and healthcare monitoring, and military surveillance. Anomaly detection strategies have yielded good outcomes in management, detecting visible light curves, identifying outliers in astronomical data, and detecting fraud in credit card transactions [6,7]. This paper will comprehensively evaluate the latest studies conducted to identify anomalies.

Wang et al. [8] examined anomaly identification in time-series data from a manufacturing system using recurrent neural networks (RNNs). This technology enabled manufacturers to detect abnormalities produced during system operation. The model’s performance was assessed in a diesel engine assembly process, successfully identifying three prevalent kinds of abnormalities using time-series data. Ibrahim et al. [9] proposed a bifurcated framework for anomaly identification in univariate time series. The first step of this investigation was time series forecasting. The second phase pertained to anomaly detection. This research used a mix of convolutional neural networks (CNNs) and bidirectional long short-term memory (LSTM) networks for the forecasting phase. Additionally, they used the mean absolute error technique throughout the detection phase. The findings of this research indicated that the suggested framework outperformed other anomaly detection approaches on the analyzed dataset, particularly regarding accuracy. Wu et al. [10] used a hybrid approach to deep learning for anomaly detection, termed reinforcement learning from pixels for autonomous driving (RLAD). The researchers used this combination to identify and adjust to real-world time-series data abnormalities. Their approach eliminates the need for manual parameter optimization. This strategy necessitated an expert to annotate the samples. The RLAD approach exhibited higher performance relative to unsupervised and semi-supervised algorithms when analogous labels were accessible. Schlegl et al. [11] introduced a novel approach for anomaly identification in medical pictures using generative adversarial networks (GANs). The researchers suggested that architecture had two components—a generator and a discriminator—that necessitated the retrieval of latent variables z using random gradient descent for each test sample during training. Nonetheless, the gradient computation in their suggested methodology required substantial processing resources and proved economically unfeasible for time series with expanding sample lengths. Akcay et al. [12] used a novel architecture using GANs for automated anomaly identification. The research demonstrated that using encoder–decoder–encoder networks in GANs facilitated the mapping of the input picture to a lower-dimensional vector, enabling the reconstruction of the output image from the corresponding vector. Nevertheless, the methodology outlined in this work was appropriate for image-based databases but exhibited suboptimal performance for time-series data. Zenati et al. [13] introduced two distinct GAN designs for anomaly identification in time data. LSTM networks were used in both generative and discriminative designs to enhance the temporal correlation of time series distributions. Also, in this study, instead of considering each data stream independently, the entire set of variables is considered simultaneously for multivariate anomaly detection to capture the latent relationships among variables. This research employs the Euclidean distance between the generator’s input and output, and downsampling is executed to mitigate computing demands. Using LSTM in the generative and discriminating architectures can improve the results to some extent. Chen et al. [14] introduced a novel paradigm for automated anomaly identification. This work integrated prediction with anomaly detection to facilitate unsupervised anomaly detection and enhance predictive robustness. This study’s suggested model used LSTM for anomaly detection. This work presented an approach whereby, during the training phase, noise and anomalies were input into the system, and the LSTM prediction block used a clean input derived from the reconstructed time series by the variational autoencoder (VAE), enhancing its robustness against anomalies and noise for predictive purposes. Qin et al. [15] introduced a novel ECG-ADGAN approach for single-class ECG classification. Their architecture integrated a BLSTM layer within GANs to maintain temporal patterns. The method demonstrated efficacy in identifying unknown anomalies within the MIT-BIH arrhythmia database. Nonetheless, it was unable to differentiate between various types of abnormalities without human annotation, in contrast to the supervised model. Liu et al. [16] examine the challenges and limitations of current metrics and datasets for anomaly detection in time-series data. The authors argue that many existing methods do not provide reliable results due to heterogeneity and poor quality of the evaluation data. They suggest that a standardized and reliable framework for evaluating anomaly detection methods should be developed to enable fairer and more meaningful comparisons. Finally, the paper emphasizes improving datasets and evaluation metrics to advance in this field. Hamon et al. [17] review unsupervised feature construction methods to improve anomaly detection in time-series data. The authors evaluate various methods for extracting meaningful features from time-series data and analyze the impact of these features on the performance of anomaly detection models. The results show that constructing appropriate features can significantly increase the accuracy and reliability of models. Finally, the paper emphasizes the importance of selecting and designing effective features to improve anomaly detection. Wang et al. [18] provide a comprehensive review of anomaly detection methods in multivariate time-series data using deep learning techniques. The authors first provide a taxonomy of existing methods, categorizing them based on neural network architectures, learning approaches, and specific applications. Then, the applications of these methods in various domains, such as equipment health monitoring, fraud detection, and industrial systems monitoring, are reviewed. Finally, the paper identifies current challenges and future research directions and provides suggestions for improving anomaly detection methods. This paper is a useful reference for researchers and practitioners in this field. Geiger et al. [19] used two generators and two discriminators for the automated detection of abnormalities. This research used the Wasserstein cost function to align the distribution of the produced time series sequences with that of the target domain data. This work used dynamic temporal warping to assess the similarity between the two time series, effectively addressing time-varying challenges. Xu et al. [20] used a time series-to-2D matrix transformation method for the automated detection of abnormalities. This model converts 1D time series into 2D pictures for input and assesses the reconstruction error to identify abnormalities in univariate series, using time series to image variational autoencoder (T2IVAE) during training to mitigate data overfitting. This approach employs a hierarchical multiscale encoder–decoder design.

A few studies have focused on employing graph deep networks to investigate anomalies and time predictions. Many studies have been based on LSTM networks, and the conventional methods are supervised. Although deep learning methods have improved anomaly detection, the information between the time samples in the previous methods has been ignored. To overcome the limitations mentioned above, a correlation-based graph embedding of time samples is constructed, and a Chebyshev graph network is proposed to recognize the anomalies. A Chebyshev graph network is an advanced method for graph data processing used in graph neural networks (GNNs). It uses Chebyshev polynomials to approximate spectral filters on graphs. This graph makes it possible to employ the distribution and the connections of the samples in the detection procedure. Furthermore, the proposed network architecture provides an unsupervised version of learning without labeling the data, and it works based on the prediction of the samples. The highlights of this study are as follows:(I)An unsupervised deep graph network is provided for the prediction of time series.(II)The network architecture performs using a graph of time samples to consider the connections in the time distribution of samples.(III)A fusion of graph networks and generative adversarial networks is proposed to improve the performance of the graph network.(IV)A modified cost function is provided for the adversarial part of the network architecture to obtain efficient weights for the network with fast convergence and achieve desired results with a low number of iterations.(V)The adversarial part works utilizing one-dimensional convolutional layers.

The remainder of the paper is structured as follows: In Section 2, the database is described, and the algorithms employed to process the data in this study are reviewed. Section 3 provides a detailed overview of the deep architecture employed in this study, as well as the proposed methodology. Section 4 presents the simulation results and evaluates the optimizations associated with the proposed architecture. Lastly, the results are addressed in Section 5.

## 2. Materials and Methods

This section comprises two subsections. The first subsection pertains to the database used, whereas the subsequent subsection analyzes the methods employed in this research.

### 2.1. Numenta Benchmark Database

Numenta anomaly benchmark (NAB) is a standard database and framework for evaluating anomaly detection algorithms on time-series data. Numenta developed the dataset, which consists of real and synthetic data used in various scenarios. NAB is used to evaluate time-series anomaly detection algorithms, which include accurate and labeled data, evaluation criteria, execution scripts, artificial intelligence, cybersecurity, the Internet of Things, and financial and medical data analytics. The dataset included in this study is a streamlined benchmark dataset often utilized in anomaly detection research [21].

This dataset has established a benchmark for assessing anomaly research for the following reasons: a. It encompasses all kinds of streaming data abnormalities. b. It encompasses a range of data measures. c. It encompasses data characterized by noise issues and emerging normative patterns.

This dataset consists of six different categories of signals, including AdExchange, KnownCause, Tweets, AWS, and artificial signals with an anomaly. The considered signals for this study are one AdExchange, two signals of KnownCause, four Tweets, three AWS signals, and four artificial signals with an anomaly. The AdExchange signal is a per-hour data record over two months (1500 samples). The first KnownCause is a record of 14 days of fluctuations every 5 min (4032 samples), and the second KnownCause is a record of 18 days of fluctuations every 5-min (5184 samples). There are four different sets of Tweets signals (AAPL, CRM, FB, and AMZN) consisting of 55 days of recordings every 5 min (15,840 samples). Finally, the three AWS signals are 14 days of recordings (4032). Likewise, each of the artificial datasets with an anomaly includes 4032 samples.

Figure 1 illustrates the signals with red samples highlighted to show the section of the signal regarding the anomalies. This figure consists of three types of artificial anomalies (jumps-up, jumps-down, no-jump), one type of AWS, one type of Tweets, and one sort of KnownCause. The number of time samples and number of anomalies corresponding to 14 types of signals can be seen in Table 1. This table explains the number of anomalies, the time samples, the train length, and the test length considered to analyze each signal of this database.

A detailed explanation of the exact location of the anomaly slices related to each signal is shown in Table 2. There are 14 types of signals with an anomaly, considering the slice locations of the anomalies.

### 2.2. Algorithms Used

This subsection delineates the methods used in this research for anomaly detection and identification. The techniques examined include GANs and convolutional graph theory. These algorithms are popular machine learning algorithms used in automation applications based on artificial intelligence. Machine learning and its subset called artificial intelligence are used today in many applications, including financial markets [22,23,24,25,26,27,28,29,30,31,32], management strategies [33], aerospace [34], deep learning algorithms [35,36,37], COVID-19 detection [38], gaming [39], and industry [40].

#### 2.2.1. GANs

Generative deep modeling can be considered as an unsupervised learning operation in which the input dataset is trained in such a way that the output model can be used to generate new examples using the original dataset. One of these generative models is the adversarial generative network, which uses two sub-networks, the generator and the discriminator, to generate synthetic data [41].

The function of the generative network entails that the generator sub-network be trained to generate new examples and the discriminator sub-network detect whether the generated example is artificial or real. In this regard, the generator should try to produce examples that are generated close enough to the real examples that the discriminator network cannot detect the artificial data generated. To this end, both sub-networks are trained simultaneously so that the generator model solves the random vector Y obtained from the prior distribution. This network is trained in a contradictory manner until none of the sub-networks can make more progress against the other. According to the following equation, *D*(*G*(*y*)) is the output of the discriminative sub-network based on the data generated by *G*(*y*)*,* considering the samples generated.(1)$$\begin{array}{l} \underset{Gen}{min}\underset{Disc}{max}V\left(D,\;G\right)=\\ \underset{Gen}{min}\underset{Disc}{max}\;\left[\begin{array}{l} E_{X\;P_{data^{\left(x\right)}}}\left[logDisc\left(x\right)\right]\\ +E_{Y\;p_{y^{\left(y\right)}}}\left[log\left(1-Disc\left(Gen\left(y\right)\right)\right)\right]? \end{array}\right]? \end{array}$$

In the above equation, *X* represents the real data and *Y* represents the feature vector imposed on *G*. Also, *G*(*y*) is the output of the generative sub-network based on the feature vector *Y*. *D*(*x*) is also related to the output of the discriminative sub-network with the real input data, an output which needs to be close to 1 for better performance. The above equation also represents the probability densities of *X* and *Y* by *P_data_*_(*x*)_ and *P*_y(y)_.

In general, it can be said that in the process of training a generative sub-network, *G* will be trained in such a way that log (1 − *D*(*G*(*y*))) decreases. This causes the discriminant sub-network to be misled, and the discriminant sub-network is trained in a way that increases the probability of the generated samples. Each GAN sub-network has a cost function that can be changed depending on its position in different domains [41].

#### 2.2.2. Convolutional Neural Network and Graph Theory

This section provides a concise overview of graph theory and graph signal processing (GSP) [42]. The functions associated with GSP take into account the characteristics of graph components and the graph’s structure. GSP extends standard convolutions for use in graph theory. Consequently, signal processing operations like the Fourier transform in GSP provide a spectrum filter in graph theory, often known as a convolutional graph.

In mathematical notation, ***D*** ∈ ℝ ^(***N***×***N***)^ denotes the diagonal degree matrix, whereas ***W*** ∈ ℝ ^(***N***×***N***)^ represents the adjacency matrix of a graph. The *i*-th diagonal element of the degree matrix is also determined using the following relation:(2)$$D_{ii}=\sum_{i}W_{ij}$$

The Laplacian matrix L is defined as follows:(3)L=D−W∈ℝN×N

The basis functions can also be computed in relation to the eigenvectors of the graph’s Laplacian matrix. The singular value decomposition can derive the eigenvectors of the Laplacian matrix (U) by the following relation:(4)$$L=U\Lambda U^{T}$$

In accordance with the aforementioned relation, the columns ***U*** = [***u***_0_, …, −1] ∈ ℝN×N signify the Fourier basis, whereas *Λ* = *d**i**a**g*([*λ*_0_, …, *λ**N* − 1]) denotes a diagonal matrix. The computation of the Laplacian eigenvectors yields the Fourier basis for the graph.

For an arbitrary signal represented by ***X*** ∈ R^***N***^, which denotes the feature vectors aggregated at the graph nodes, the Fourier transform of the graph’s GFT is derived using the graph’s basis functions as follows:(5)$$\hat{x}=U^{T}x$$

In the aforementioned connection, x denotes the converted signal in the frequency domain, expressed as the solution of the Fourier transform of the graph. The inverse of GFT can alternatively be defined as follows:(6)$$x=UU^{T}x=U\hat{x}$$

The complexity of two signals in the graph domain is determined for convolutional graph operations using GFT. The intricacy of ***x*** and ***y*** in the graph ∗***f*** can be articulated as follows:(7)x( g∗)Y=U(((UT)x)⊙((UT)Y))

In the aforementioned connection, the symbol ⊙ denotes the Hadamard product, computed for the graph’s Fourier transform signals. To establish a filter function predicated on (), one may proceed as follows:(8)y=f(L)x

The aforementioned relationship may be encapsulated as follows:(9)$$\begin{array}{l} y=f(L)x=Uf(\Lambda )U^{T}x=U(f(\Lambda )).(U^{T}x)\\ =U(U^{T}(Uf(\Lambda ))).(U^{T}x)={x^{*}}_{f}(Uf(\Lambda )) \end{array}$$
where(10)$$f(\Lambda )=\left[\begin{array}{lll} f(\lambda_{0}) & \cdots & 0\\ \vdots & \ddots & \vdots \\ 0 & \cdots & f(\lambda_{N-1}) \end{array}\right]$$

This research employs a specific variant of the convolutional graph [43]. Consequently, in this context, *f*(***L***) is substituted with the Chebyshev polynomial expansion L. The complexity of the graph X may be described using ***U****f*(***Λ***) as follows:(11)$$y=f(L)x=f(U\Lambda U^{T})x=Uf(\Lambda )U^{T}x$$

L is derived from W by relation 2, and ***Λ*** can be computed using relation 3. The function *f*(***Λ***) may be computed using Chebyshev polynomials, whereas the maximum value among the diagonal elements of ***Λ*** is ascertained by *λ**M**a**x*. Consequently, the normalized ***Λ*** can be articulated as follows:(12)$$\lambda =2\Lambda /\lambda_{\max}-I_{N}$$

In the above relation, i denotes the identity matrix of dimensions x by x, and the diagonal components corresponding to $\tilde{\Lambda }$ lie inside the range [1, −1]. The differentiation of (***Λ***) can alternatively be analyzed as follows:(13)$$D_{ii}=\sum_{i}W_{ij}$$

In the aforementioned connection, ***θ****k* denotes the coefficient of the Chebyshev polynomial. Furthermore, *k* ($\tilde{\Lambda }$) is derived as follows:(14)$$f(\Lambda )=\sum_{k=0}^{K-1}\theta_{k}T_{k}(\tilde{\Lambda })$$

The convolutional operation of the graph may be articulated as follows:(15)$$\left\{\begin{array}{l} T_{0}\left(\tilde{\Lambda }\right)=1,\;T_{1}\left(\tilde{\Lambda }\right)=\tilde{\Lambda }\\ T_{k}\left(\tilde{\Lambda }\right)=2(\tilde{\Lambda })(T_{k-1})(\tilde{\Lambda })-T_{k-2}(\tilde{\Lambda }) \end{array}\right.$$

In the aforementioned equation, $\hat{L}$= 2(***L***/*λ**m**a**x*) − ***I***_***N***_ denotes the normalized Laplacian matrix. The Chebyshev graph convolution indicates that it is equal to the convolution results of ***x*** with each Chebyshev polynomial component [43].(16)$$\begin{array}{l} y=U\;f(\Lambda )\;U^{T}x\\ =\sum_{k=0}^{K}U\left[\begin{array}{l} \theta_{k}T_{k}({\tilde{\lambda }}_{0})\;\cdots \;0\\ \;\vdots \;\ddots \;\vdots \\ 0\;\cdots \;\theta_{k}T_{k}({\tilde{\lambda }}_{N-1}) \end{array}\right]\;U^{T}x\\ =\sum_{k=0}^{K}\theta_{k}T_{k}(\tilde{L})x \end{array}$$

The graph Laplacian is an important matrix in graph theory and machine learning on graphs that represents the structure of a graph and is used in spectral analysis of graphs. It plays a key role in many graph processing algorithms, including GNNs, spectral clustering, and information diffusion modeling.

## 3. Proposed Model

This part will provide the suggested approach for anomaly detection, including data pre-processing, the proposed architecture, and the training and assessment set. The schematic overview of the proposed method is represented in Figure 2.

### 3.1. Pre-Processing Stage

The Numenta signals are considered, and the normalization is performed to scale the parameters to a specific domain of (−1,1) [44]. The next step is the windowing to subsample the time series for processing [45]. A specific window size is considered to divide the signal into subsamples and train the network according to the extracted subsamples. The stride step of the window is considered equal to 1. The stacked outcome of the windowing stage is imposed onto the graph construction phase. The target for training the unsupervised network would be the first time sample after the last sample in the window.

### 3.2. Graph Construction

After pre-processing, the graph construction stage is required to impose the constructed graph onto the proposed network architecture. Ten consecutive frames of the sub-sampled signal are required for graph embedding. Cross-correlation between these consecutive signals is calculated to obtain the matrix of cross-correlation. A sigmoid is employed for calculating the absolute value of the cross-correlation matrix. The adjacency matrix is the output of the sigmoid function according to the schematic overview of the graph construction stage in Figure 2.

### 3.3. Proposed Chebyshev Convolution-Based Modified Adversarial Network (Cheb-MA)

The schematic overview of the proposed method is represented in Figure 3. The Numenta signals are used as the database, and there are sections of signals with an anomaly that can be detected through this procedure. As can be seen in this figure, after the pre-processing and graph construction stage, the obtained graph would be imposed onto the proposed Cheb-MA to perform the training. The Cheb-MA consists of two parts of deep networks. The Cheb part consists of three Chebyshev convolution layers, and the output vector is considered to train the generator and discriminator. The training phase of the Cheb-MA is completed with the parts of the signal without an anomaly. The signal with an anomaly in the test phase would be detected with the trained generator and discriminator. The anomaly predictions are according to the predictions of the discriminator.

### 3.4. Proposed Cheb-MA Architecture

The complete schematic of the suggested network design is shown in Figure 4. Cheb-MA incorporates three graph convolutional layers, as this image illustrates. This graphic indicates that estimating the Chebyshev convolution of the input graph using the graph Laplacian is the first step in each Chebyshev convolutional layer. Each layer’s activation function is the rectified linear unit (ReLU). Each layer’s output is then placed on the next layer after passing through a batch normalization filter.

Batch normalization helps the network become more stable during the training procedure and makes the training process converge more quickly. This normalization is performed after the Chebnets. After three layers of Chebyshev convolution, the extracted feature vector is flattened and passed through two dense layers to be compatible with the target size. The details of different parts of the proposed architecture are explained in Table 3 and are related to the details of the first part of the Cheb-MA. According to Table 3, as is clear, the number of layers used in this architecture is seven. Also, the layers used in this section include Chebyshev convolution layers, batch normalization, and a dense layer, which are used through the ReLU activation function. Table 4 shows the characteristics of layers corresponding to the generator part, and Table 5 is concerned with the discriminator. Table 4 delineates the specifications of the transposed one-dimensional convolution. According to Table 4, as is clear, the weight parameter has decreased from 2000 to 4 from the initial to the final layer, respectively. Also, as is clear, the alpha parameter in the activation function has been set to 0.1. The dimensions related to the outputs of the one-dimensional convolutions in the discriminator can be seen in Table 5. Also, it shows the kernel size, the size of strides, the number of kernels in each layer, and the number of parameters to be trained during the training procedure. As is clear, the model consists of eight different layers, the initial layer starting with the Chebyshev convolution layer and ending with the dense layer.

### 3.5. Training and Evaluation of the Unsupervised Cheb-MA

In the training phase, the input and target samples are created according to Figure 5. Input time series for the Chebyshev convolution network of the Cheb-MA are {T0… Td}, {T1…T1 + d}, {T2… T2 + d}, …, {Tn…Tn + d}, and the corresponding targets are Td + 1, Td + 2, Td + 3, …, Td + n + 1. The input time series for the GAN part of the proposed network are of the form {T0 + 1… Td + 1}, {T2…T2 + d}, {T3… T3 + d}, …., {Tn…Tn + d}, and the corresponding targets are {T0… Td}, {T1…T1 + d}, {T2… T2 + d}, …., {Tn − 1…Tn − 1 + d}.

In order to tune the weights of the proposed Cheb-MA to the Numenta database, we implement a 10-fold cross-validation for the section without an anomaly. The testing phase is performed after training and tuning the weights of the Chebyshev convolution network and the adversarial network.

The training of the proposed unsupervised Cheb-MA according to the parameters in Table 6 is performed. The optimal parameters are acquired and illustrated in this table. Cross-validation [46] is required, and the procedure is shown in Figure 6.

According to Figure 6, 10-fold cross-validation is performed using the training sections of the signal without an anomaly. The test stage can predict the anomalies based on the obtained weights of the training step. The proposed model in Algorithm 1 has been examined in detail.
**Algorithm 1 Proposed method.**Chebyshev convolution-based modified adversarial network (Cheb-MA) Input: (1) Multichannel EEG signal ***X***. (2) Window size, striding size, α (modifying coefficient of the adversarial network).(3) Coefficient for modified adversarial network. (4) Train and Test Sequence ***X******t******r******a******i******n*** and ***X******t******e******s******t***. Output: Train Target and Test TargetInitialize the model parameters. Repeat according to the 10-fold cross-validation: 1: Compute the correlation coefficient of the ***X*** in ***X******t******r******a******i******n***. 2: Obtain the adjacency matrix ***W*** via using the sigmoid function for the result of step 1.3: Compute the normalized Laplacian matrix Λ^.4: Calculate the Chebyshev polynomials. 5: Compute the output of the three Chebyshev convolution layers and regularize using the Relu operator. 6: Calculate the Dense layer. 7: Update the weights of the layers using cross-entropy as loss function. 8: Impose the result onto the generator and discriminator part. 9: Update the weights of the layers using modified cost function: Generator:LossBinary−Cross−Entropy(1,fake) = − 1n ∑i=1n(1.log(fakei)+(1−1).log(1− fakei)) =− 1n ∑i=1n(1.log(fakei)) Discriminator:LossBinary−Cross−Entropy(α∗1, real) = − 1n ∑i=1n(α∗1.logreali+(1−α∗1).log(1−reali))LossBinary−Cross−Entropy(0,fake) = − 1n ∑i=1n(0.logfakei+(1−0).log(1−fakei)) = − 1n ∑i=1nlog(1−fakei)LossTotal= LossBinary−Cross−Entropy(α∗1, real) + LossBinary−Cross−Entropy(0,fake)  10: Calculate the predictions for the constructed graphs according to ***X******t******e******s******t*** using the trained Cheb-MA.  Stop criterion: Until either a maximum number of iterations or efficient accuracy is acquired.

## 4. Results and Discussion

The simulation results of the suggested Cheb-MA are shown in this section. Our framework runs on a laptop equipped with a GeForce GTX 1050 GPU, a 2.8 GHz Core i7 CPU, and 16 GB of RAM. The Google Colaboratory Pro platform is used to train the suggested network.

Figure 7 shows the performance of the proposed Cheb-MA based on the loss functions according to the Chebyshev convolution part, generator, and discriminator section of the network. According to this figure, the SGD optimizer has a learning rate of 0.0001 and weight decay of 4 × (10^−4^), considering cross-entropy for the first part of the network and the modified total loss according to Algorithm 1 for the adversarial part of the proposed network. Also, to clarify the performance improvement in the adversarial generative network, we have compared the loss function of the proposed discriminator with the classical discriminator in Figure 8. As is clear, the proposed improved discriminator has a more stable performance compared to its counterpart. Figure 9 demonstrates the fluctuations of the loss function corresponding to the MA network architecture for unsupervised anomaly detection. The GNN and LSTM loss function variations can be seen in Figure 10. The GNN consists of three layers of graph convolution networks, and the LSTM network includes four stacked layers of LSTM with 64 units for the first three layers and 1 unit for the last LSTM layer.

As can be seen, we consider 40 iterations for LSTM, GAN, and GNN methods and 10 iterations for each fold of the 10-fold cross-validation. As a result, we have 400 iterations for the loss functions. For the convergence of our proposed method, more iterations are required. The convergence of the proposed Cheb-MA happens after 700 iterations (70 iterations of 10-fold cross-validation according to the 10 training repetition of each fold).

The results of the proposed Cheb-MA together with LSTM networks, GNNs, and GANs are available in Table 7. Moreover, this table presents the class-specific performance evaluation metrics in detail for all of these networks. As can be seen in Table 7, the proposed Cheb-MA outperforms the other methods. All of the evaluation metrics for anomaly detection work based on the TP, TF, FP, and FN. The true detections of anomalies are TP, and false detections are FP. The anomalies that cannot be detected are FN, and the true detections according to normal sections are TN. We note that in all experimental settings of the proposed model, the various features and parameters used in the research being compared follow a comprehensive standard. The NAB score [47] is another evaluation metric for the Numenta anomaly benchmark dataset. It considers a window profile for positively rewarding the true positives, penalizing the false negatives and the remote distance false positives. The judgement of this score does not differ from the F1-score in terms of true negatives. Table 8 compares the results of this evaluation metric for 14 signals of the Numenta database. Table 9 shows the performance of deep learning methods in comparison to the conventional methods in terms of the NAB score. As can be seen, deep learning approaches outperform the other conventional methods. Table 10 shows the comparison of the F1-score with the state-of-the-art deep learning methods.

To elaborate more on the effect of different parameters, we conduct a separate experiment. In order to assess the effect of changing the number of Chebyshev layers, a series of simulations are carried out for different numbers of Chebyshev polynomials. Figure 11 demonstrates the results of the simulation for two, three, four, and five Chebyshev layers. Increasing the layers to more than three in this case study does not affect the performance; it causes a computational burden, and it shows the incremental trend of the training time per epoch of the proposed Cheb-MA. Figure 12 shows the comparison result of different modifying coefficients of the adversarial part of the network. This column chart delineates that setting the coefficient equal to 0.9 will improve the F1-score, and it is the efficient one considering the time of convergence. The confusion matrix in accordance with the four types of Numenta signals is illustrated in Figure 13.

A paired T-test in Table 11 is performed to validate the accuracy of Cheb-MA compared to GNN and LSTM. As can be seen, the performance of Cheb-MA is better than other methods, considering different ranges for *p*-value. This shows the strength of the proposed Cheb-MA with T-Test cross-validation, considering five tests.

In this study, an unsupervised deep graph network was presented for time series decay. The proposed network architecture was formed using a graph of time samples to consider connections in the time distribution of samples. The proposed architecture included a combination of graph networks and GANs to improve the performance of the graph network, in which a modified cost function was used for the adversarial part of the network architecture. As it was examined, our proposed architecture was also tested with various networks, including classical GAN, LSTM, and manual learning methods, and demonstrated the best performance. Also, the proposed study was compared in standard conditions based on the same database with studies [18,45,48,49,50] in terms of the F1-score evaluation index and was able to demonstrate promising results. It is recommended to consider another graph construction technique, such as sparse graph representation, to decrease the computational burden. Also, another network of generators and discriminators consisting of graph convolutional networks can be proposed in future works to improve the accuracies of the anomaly detection procedure.

## 5. Conclusions

In this study, a novel fusion strategy is proposed to detect anomalies and predict time series in an unsupervised way. The correlation between the neighboring time samples of data fluctuations is used for graph construction. The prediction of the anomalies based on the graph representation is realized utilizing a deep network consisting of Chebyshev convolutions and a one-dimensional convolution-based generative adversarial network. Furthermore, the suggested modification of cost function adjusts the weights of the adversarial network to attain a desired performance. Employing the Chebyshev convolution along with the modified adversarial network delineates the efficiency of the proposed method. Higher evaluation metrics in terms of F1-score and Numenta anomaly benchmark score are achieved compared to other state-of-the art deep learning methods.

## Figures and Tables

**Figure 1 biomimetics-10-00245-f001:**
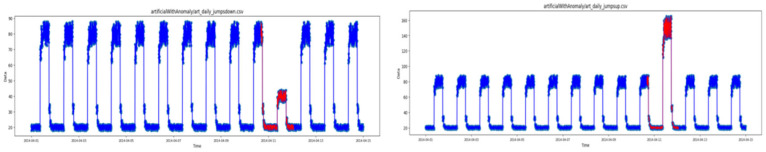
Numenta database signal samples (Blue: Normal samples, red: Anomaly).

**Figure 2 biomimetics-10-00245-f002:**
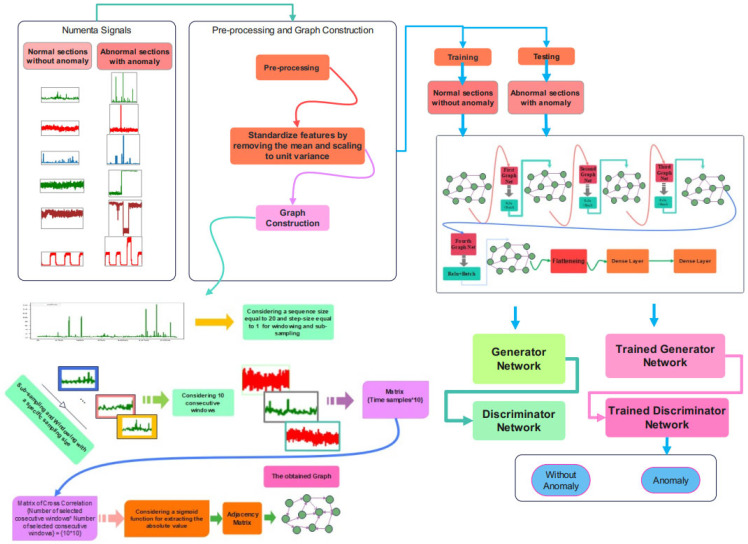
A schematic overview of the proposed anomaly detection.

**Figure 3 biomimetics-10-00245-f003:**
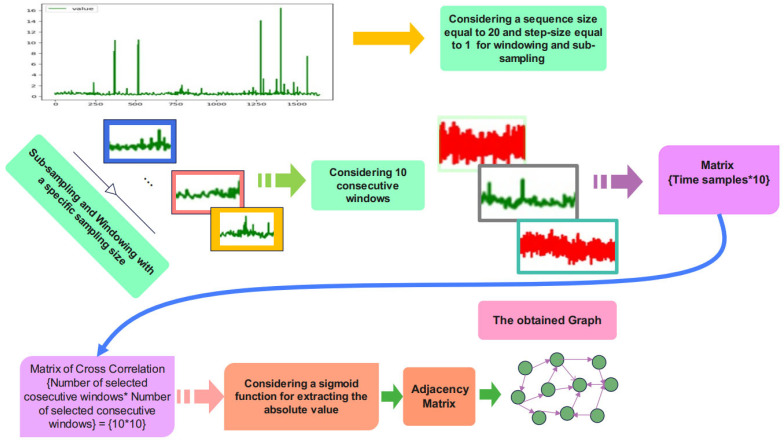
Graph construction.

**Figure 4 biomimetics-10-00245-f004:**
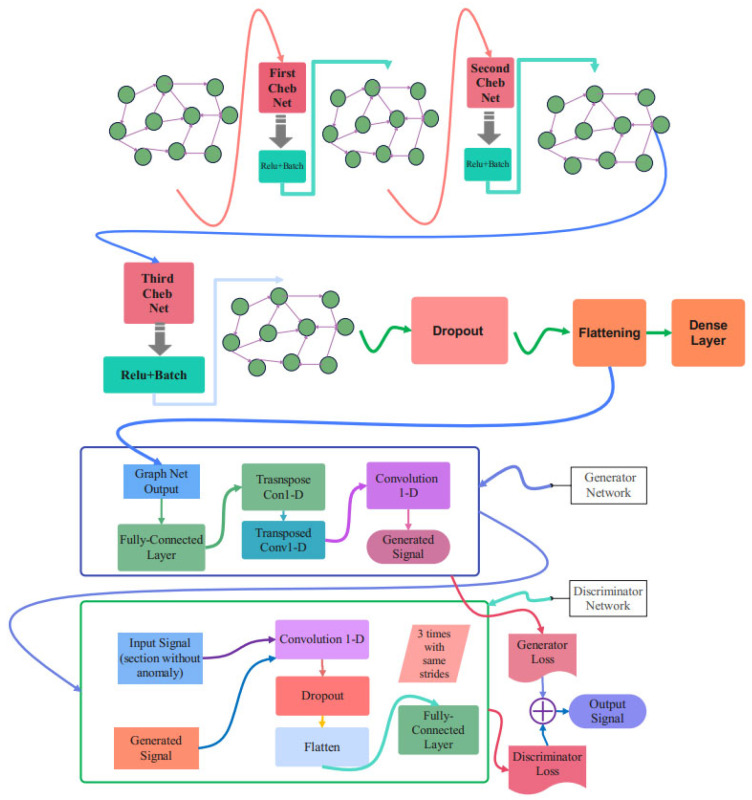
The proposed network architecture.

**Figure 5 biomimetics-10-00245-f005:**
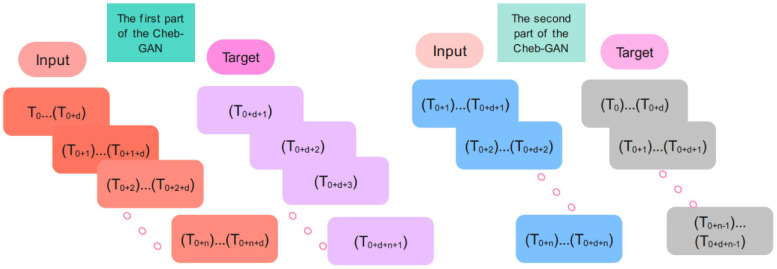
The input and targets for the training procedure.

**Figure 6 biomimetics-10-00245-f006:**
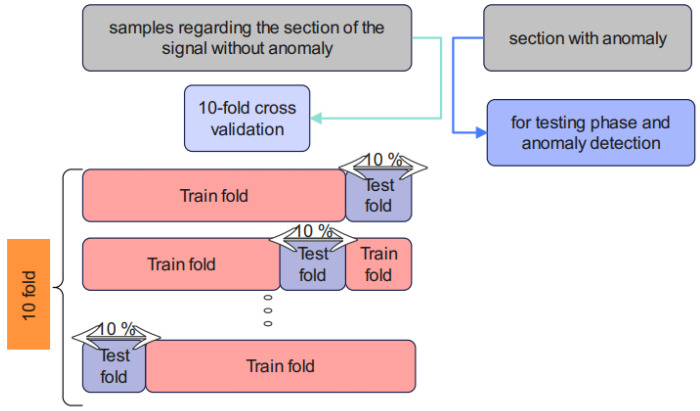
K-fold cross-validation of the proposed method.

**Figure 7 biomimetics-10-00245-f007:**
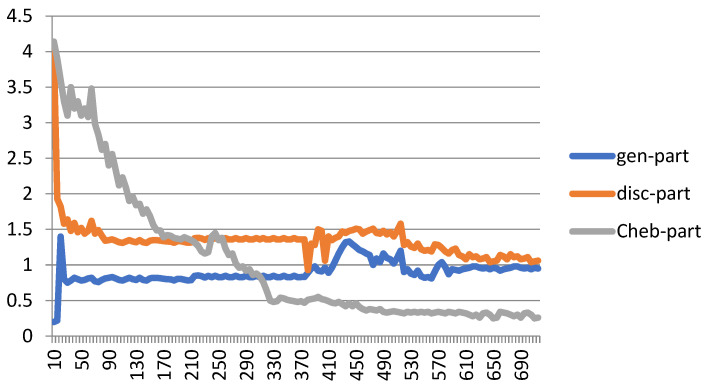
Loss plot for training the proposed Cheb-MA architecture.

**Figure 8 biomimetics-10-00245-f008:**
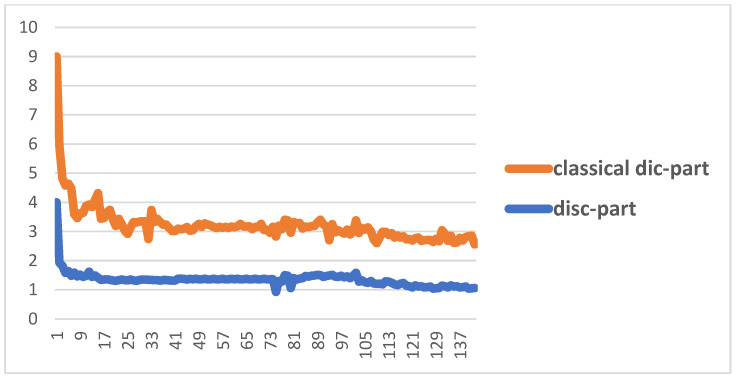
Comparison between the improved discriminator and the classic discriminator.

**Figure 9 biomimetics-10-00245-f009:**
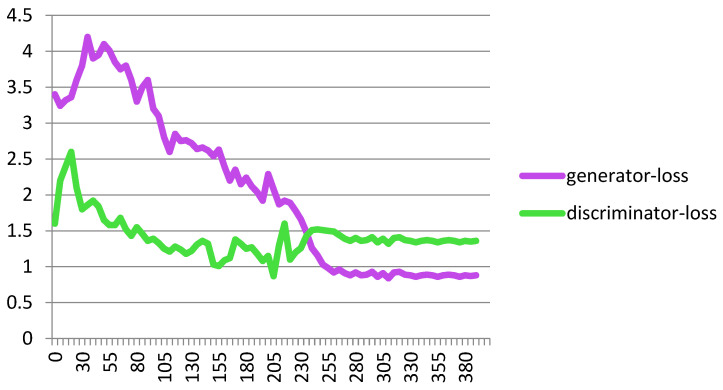
Loss plot for the GAN.

**Figure 10 biomimetics-10-00245-f010:**
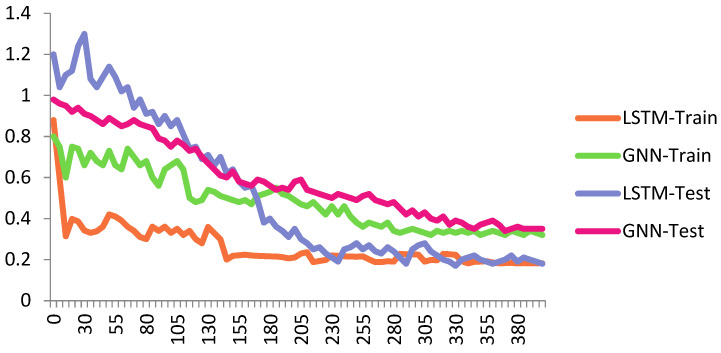
Loss plot for GNN and LSTM train and test sets.

**Figure 11 biomimetics-10-00245-f011:**
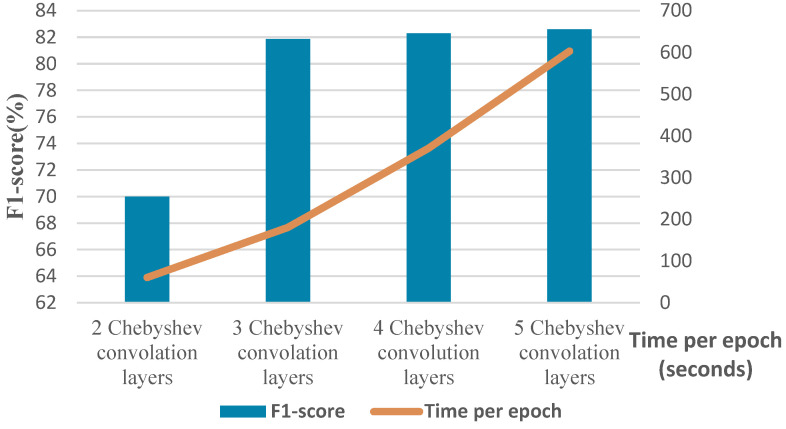
F1-score and time of training per epoch with different numbers of graph convolution layers.

**Figure 12 biomimetics-10-00245-f012:**
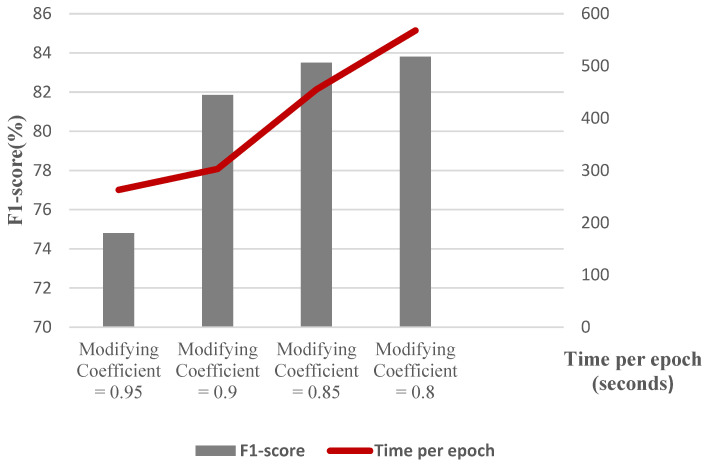
F1-score and training time per epoch for graph construction with varying sparsities.

**Figure 13 biomimetics-10-00245-f013:**
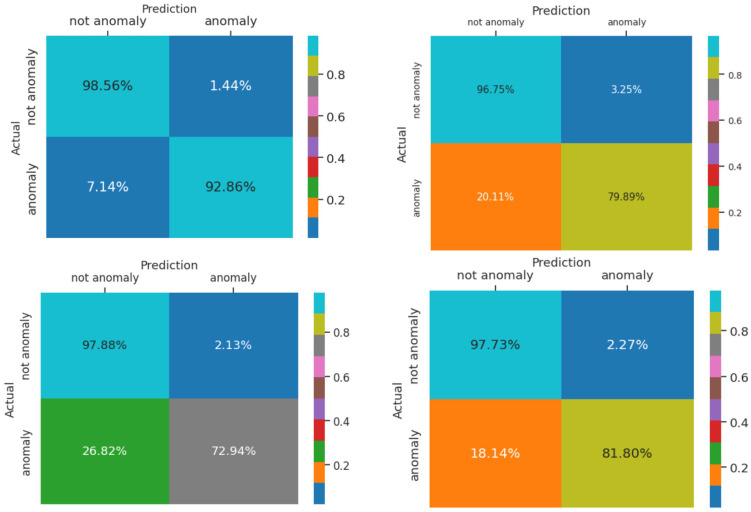
Confusion matrix for Art1, AdEx1, AWS1, and Tweet 1 with Cheb-MA.

**Table 1 biomimetics-10-00245-t001:** Numenta-related specifications.

Numenta	Category	Subset	Time Sample	Number of Anomalies	Train Length	Test Length
1	AdExchange	AdEx1	1600	184	1000	600
2	KnownCause	Cause1	5184	530	4000	1200
3	KnownCause	Cause2	4032	346	3000	1000
4	Tweets	Twitter1	15,840	1582	10,000	5840
5	Tweets	Twitter2	15,840	1588	10,000	5840
6	Tweets	Twitter3	15,840	1580	10,000	5840
7	Tweets	Twitter4	15,840	1593	10,000	5840
8	AWS	AWS1	4032	343	3000	1032
9	AWS	AWS2	4032	403	3000	1032
10	AWS	AWS3	4032	403	3000	1032
11	Artificial	Art1	4032	403	3000	1032
12	Artificial	Art2	4032	403	3000	1032
13	Artificial	Art3	4032	403	3000	1032
14	Artificial	Art4	4032	403	3000	1032

**Table 2 biomimetics-10-00245-t002:** Details about the database used.

Numenta	Category	Subset	Anomaly Slice	Numenta	Category	Subset	Anomaly Slice
1	AdExchange	AdEx1	Slice (347,380) Slice (729,769) Slice (1256,1296) Slice (1381,1421)	8	AWS	AWS1	Slice (1526,1868)
2	Known Cause	Cause1	Slice (2243,2507) Slice (2877,3141)	9	AWS	AWS2	Slice (1437,1839)
3	Known Cause	Cause2	Slice (2014,2148) Slice (3328,3462) Slice (3956,4031)	10	AWS	AWS3	Slice (3374,3776)
4	Tweets	Twitter1	Slice (4614,5404) Slice (9926,10,716)	11	Artificial	Art1	Slice (2679,3081)
5	Tweets	Twitter2	Slice (1235,1631) Slice (2920,3316) Slice (4761,5157) Slice (9087,9483)	12	Artificial	Art2	Slice (2787,3189)
6	Tweets	Twitter3	Slice (1796,2190) Slice (3538,3932) Slice (9598,9992) Slice (11,409,11,803)	13	Artificial	Art3	Slice (2787,3189)
7	Tweets	Twitter4	Slice (2872,3402) Slice (5800,6330) Slice (7768,8298)	14	Artificial	Art4	Slice (2787,3189)

**Table 3 biomimetics-10-00245-t003:** Numenta anomaly specifications and slices.

Layer	Layer Name	Activation Function	Dimension of Weight Array	Dimension of Bias	Number of Parameters
1	Chebyshev convolution layer	ReLU	[1, 20, 20]	[20]	420
2	Batch normalization	-	[20]	[20]	40
3	Chebyshev convolution layer	ReLU	[1, 20, 20]	[20]	420
4	Batch normalization	-	[20]	[20]	40
5	Chebyshev convolution layer	ReLU	[1, 20, 10]	[5]	205
6	Batch normalization	-	[10]	[10]	20
7	Dense layer	-	[100, 1]	[1]	101

**Table 4 biomimetics-10-00245-t004:** Information about the Cheb-MA’s generating component.

Layer	Layer Name	Activation Function	Output Dimension	Size of Kernel	Stride Shape	Number of Kernels	Number of Weights
2	Dense	LeakyReLU (alpha = 0.1)	(1, 20)				2000
3	Reshape		(1, 20,1)				0
4	Transposed Convolution 1-D	LeakyReLU (alpha = 0.1)	(1, 20, 8)	1 × 4	1 × 1	8	256
5	Transposed Convolution 1-D	LeakyReLU (alpha = 0.1)	(1, 20, 8)	1 × 4	1 × 1	8	256
6	Transposed Convolution 1-D	LeakyReLU (alpha = 0.1)	(1, 20, 1)	1 × 4	1 × 1	1	4

**Table 5 biomimetics-10-00245-t005:** Details on the Cheb-MA’s discriminator component.

Layer	Layer Name	Activation Function	Output Dimension	Size of Kernel	Strides	Number of Kernels	Number of Weights
1	Convolution 1-D	LeakyReLU (alpha = 0.1)	(1, 20, 4)	1 × 4	1 × 1	4	20
2	Dropout (0.3)		(1, 20,4)				0
3	Convolution 1-D	LeakyReLU (alpha = 0.1)	(1, 10, 4)	1 × 4	1 × 1	4	68
4	Dropout (0.3)		(1, 10,4)				0
5	Convolution 1-D	LeakyReLU (alpha = 0.1)	(1, 5, 4)	1 × 4	1 × 1	4	68
6	Dropout (0.3)		(1, 5, 4)				0
7	Flatten		(1, 20)				0
8	Dense		(1, 1)				21

**Table 6 biomimetics-10-00245-t006:** Details of training parameters.

Parameters	Search Scope	Optimal Value
Optimizer of first part	Adam, SGD	SGD
Cost function of first part	MSE, Cross-Entropy	Cross-Entropy
Number of Chebyshev convolutional layers	2, 3, 4	3
Learning rate of first part	0.1, 0.01, 0.001	0.001
Window size	15, 20, 25, 30	20
Optimizer of MA	Adam, SGD	Adam
Learning rate of MA	0.01, 0.001, 0.0001, 0.00001	0.0001
Number of transposed 1D-convolution layers of generator of MA	2, 3, 4	4
Number of 1D-convolution layers of discriminator of MA part	2, 3, 4	3

**Table 7 biomimetics-10-00245-t007:** Performance metrics of the proposed method (accuracy, precision, recall, F1-score).

Dataset	Cheb-MA				GNN				LSTM			
	Accuracy	Precision	Recall	F1	Accuracy	Precision	Recall	F1	Accuracy	Precision	Recall	F1
**Art 1**	87.08	88	93	90	92.01	85.9	91.31	88	84.32	84.36	93.3	88.6
**Art 2**	86.2	85.3	86.4	85.84	85.36	82.06	85.4	83.69	85.23	81	83.4	82.3
**Art 3**	90.09	87	88.2	87.59	89	84.36	85	84.67	89	82	82.65	82.32
**Art 4**	82.2	64.69	92.3	76.06	80.2	64	91.31	75.48	78.6	63.2	90.02	73.65
**AdEx 1**	89.37	79.5	76	78	78	72.9	86.7	79.11	75.3	67	77.2	71.64
**AWS 1**	95.27	76.2	73	74.46	93	73	68	70	94.22	75	69.3	71.87
**AWS 2**	96.8	78.66	86	82.2	91.3	74.9	84.5	79.33	90.89	78	83	80.4
**Cause 1**	91.64	78.66	86	82.2	91.3	78	83	80.4	90.77	74.9	84.5	79.33
**Cause 2**	91.51	74	70	71.9	90.23	69.3	75	71.87	90.46	68	73	70
**Tweet 1**	94.06	80.4	82.8	80.98	93.88	73.11	85.9	78.96	93.09	72.90	88.4	79.2
**Tweet 2**	94.23	93.2	91.2	91.89	93.52	87.5	73.5	79.3	92.8	85.6	69.3	76.1
**Tweet 3**	94.02	90.2	90.82	90.51	93.2	77	91	83.4	92.75	72.04	84.8	77.5
**Tweet 4**	92.8	74.3	76.9	75.6	91.62	75	79.6	77.2	91.06	73.4	74.6	73.8
**Average**	91.17	80.77	84.04	82.09	89.4	76.69	83.09	79.33	88.34	75.18	81.03	77.43

**Table 8 biomimetics-10-00245-t008:** NAB scores of the proposed method.

Dataset	Cheb-MA	GNN	LSTM	Average Cheb-MA (%)	Average GNN (%)	Average LSTM (%)
Art 1	81.38	76.42	76.42	75.26	71.82	71.32
Art 2	81.3	79.2	79.3			
Art 3	79.6	75.3	74.23			
Art 4	58.77	56.36	55.36			
AdEx 1	80.43	67.8	62.9	80.43	67.8	62.9
AWS 1	80.49	80.4	79.33	78.54	78.5	75.51
AWS 2	76.6	76.6	71.7			
Cause 1	71.09	68.20	65.3	80.82	77.1	72.74
Cause 2	90.56	86.79	80.18			
Tweet 1	82.5	81.3	79.7	83.59	73.84	70.03
Tweet 2	96.8	76.66	70.6			
Tweet 3	75.2	68.8	62.53			
Tweet 4	79.89	68.6	67.3			
Average				79.72	73.81	70.5

**Table 9 biomimetics-10-00245-t009:** Comparison with the conventional methods.

Method	NAB Score (%)
Cheb-MA	79.72
GNN	73.8
LSTM [48]	70.5
ARTime [18]	74.9
Numenta HTM [45]	70.5–69.7
KNN CAD [49]	58.0
Random Cut Forest [50]	51.7

**Table 10 biomimetics-10-00245-t010:** Comparison with the traditional approaches.

Method	F1-Score
Cheb-MA	82.09
GNN	79.33
LSTM [48]	77.43
TadGAN [16]	70.2
Arima [51]	60.9
DeepAR [52]	56.8

**Table 11 biomimetics-10-00245-t011:** T-test evaluation to confirm the results obtained.

	Test 1	Test 2	Test 3	Test 4	Test 5
Cheb-MA	93.17%	89.42%	92.52%	92.46%	88.28%
GNN	89.4%	88.56%	89.84%	87.23%	91.97%
LSTM	87.23%	88.56%	87.43%	89.12%	89.36%
Cheb-MA, GNN	3.77	0.86	2.68	5.23	3.69
t	T = (5^0.5^) × 3.246/1.61 = 4.50				
	*p*-value < 0.02	*p*-value < 0.05	*p*-value < 0.1	*p*-value < 0.15	*p*-value < 0.2
	4.5 > 2.77	4.5 > 2.13	4.5 > 1.53	4.5 > 1.19	4.5 > 0.94
Cheb-MA, LSTM	5.94	0.86	5.09	3.34	1.08
t	t = (5^0.5^ × 3.26)/2.28 = 3.19				
	*p*-value < 0.02	*p*-value < 0.05	*p*-value < 0.1	*p*-value < 0.15	*p*-value < 0.2
	3.19 > 2.77	3.19 > 2.13	3.19 > 1.53	3.19 > 1.19	3.19 > 0.94

## Data Availability

The Numenta database is publicly available for download from the following link: https://github.com/numenta/NAB (accessed on 16 January 2025).

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
