# Peer review of "An Unsupervised Fusion Strategy for Anomaly Detection via Chebyshev Graph Convolution and a Modified Adversarial Network"

_biomimetics, 2025, doi:10.3390/biomimetics10040245_

Round 1
Reviewer 1 Report
Comments and Suggestions for Authors
In this manuscript, authors proposed "An Unsupervised Fusion Strategy for Anomaly Detection via Chebyshev Graph Convolution and Modified Adversarial Network" to predict temporal information in time series data. There are many concerns related to the contribution of this paper; its structure, writing style, and the whole presentation are summarized as follows:
1. The abstract has the following issues:
- It lacks a clear problem definition and motivation to conduct the study.
- It also lacks sufficient context or explanation for technical terms like Chebyshev graph convolution, adversarial networks, and bionics.
- This makes it difficult to understand the argument being presented.
- The methods and argument are incoherent, and the argument regarding the F1 score of 82.09%" results is not contextualized.
2. The introduction does not sufficiently explain the value of the research and how it fulfills reported gaps in anomaly detection. The barrier between the problem and the proposed solution could use more development and justify the reason behind the concern. The connection between bionics and anomaly detection is weak, and there is a lack of discussion on how bionics would be incorporated into the method.
3. The paper uses unlabelled terminology, such as Chebyshev graph convolution, graph Laplacian, and adversarial cost functions, which can disconnect readers who are not familiar with the terms. The discussion on the benchmark dataset, Numenta, leaves a significant gap in understanding.
4. The contribution of the paper is highlighted at the end but is not given the attention it deserves in the text itself. The writing is dense and can sometimes be challenging to digest due to weak transitions between sections. The paper does not manage background information, corresponding work, and methodology well.
5. The methodology is not detailed, and the figures and tables fail to effectively illustrate points. The computational complexity of the method is discussed, but not as thoroughly as required. Including Chebyshev graph convolution layers together with the adversarial network may escalate complexity, but a detailed training and inference comparison is lacking.
The writing style needs to be improved. A comperhensive professional English editing is required.
Author Response
#Reviewer 1.
Comments:
In this manuscript, authors proposed "An Unsupervised Fusion Strategy for Anomaly Detection via Chebyshev Graph Convolution and Modified Adversarial Network" to predict temporal information in time series data. There are many concerns related to the contribution of this paper; its structure, writing style, and the whole presentation are summarized as follows:
Reply: While thanking the esteemed reviewer for a thorough review of the manuscript version. We, the authors of the article, believe that your suggestions have been very useful and effective in improving the scientific version of the manuscript. We carefully answered all the questions and suggestions of the esteemed reviewer and added them to the manuscript version.
- abstract has the following issues:
Reply: Yes, the respected referee's opinion is absolutely correct. Accordingly, based on the comments below, the abstract has been rewritten as follows to resolve the relevant issues according to the respected referee's taste.
- It lacks a clear problem definition and motivation to conduct the study.
Reply: “Anomalies refer to data that is inconsistent with the overall trend of the data set and may indicate an error, or an unusual event. Time series prediction can be used to detect the anomalies which happen unexpectedly in critical situations during usage of a system or a network. Detecting or predicting anomalies in the traditional way is time-consuming and error-prone. Accordingly, the automatic recognition of the anomalies is applicable to reduce the cost of defects and will pave the way for companies to optimize their performance.”
- It also lacks sufficient context or explanation for technical terms like Chebyshev graph convolution, adversarial networks, and bionics.
Reply: “Thus, the proposed method is based on the combination of generative adversarial networks (GANs) and Chebyshev graph, which has shown good results in various domains. Accordingly, the performance of the proposed fusion approach of Chebyshev graph-based modified adversarial network (Cheb-MA) is evaluated on the Numenta dataset.”
The concept and connection of bionics with anomaly was removed from the abstract and made into an introduction, because it would confuse readers with the amount of content in the abstract.
- This makes it difficult to understand the argument being presented.
Reply: “The proposed model was evaluated based on various evaluation indices, including the average F1-score, and was able to reach a value of 82.09%%, which is very promising compared to recent research.”
- The methods and argument are incoherent, and the argument regarding the F1 score of 82.09%" results is not contextualized.
Reply: The proposed model was evaluated based on various evaluation indices, including the average F1-score, and was able to reach a value of 82.09%%, which is very promising compared to recent research.
which are highlighted on page 1, lines 8-26.
- The introduction does not sufficiently explain the value of the research and how it fulfills reported gaps in anomaly detection. The barrier between the problem and the proposed solution could use more development and justify the reason behind the concern. The connection between bionics and anomaly detection is weak, and there is a lack of discussion on how bionics would be incorporated into the method.
Reply: Yes, the respected referee's opinion is absolutely correct. In this regard, we have added the connection between anomaly and bionics to the manuscript. Also, we have made the problems that exist in diagnosing the anomaly more complete in the manuscript:
“The combination of bionics and anomaly can create interesting and new concepts in various sciences, especially in fields such as medical engineering, artificial intelligence, and biology. This combination suggests a future in which bionic systems can intelligently adapt to different conditions and identify and correct unexpected problems [3].”
“In those days, anomaly detection was done completely manually and visually by specialists and experts in each field. However, manual detection was accompanied by problems. Among the problems of manual detection can be considered human error, high detection time, uncertainty, etc. Based on the above, in recent years, automatic anomaly detection has been developed using machine learning techniques.”
which are highlighted on page 2, lines 51-55 and 66-71.
- The paper uses unlabelled terminology, such as Chebyshev graph convolution, graph Laplacian, and adversarial cost functions, which can disconnect readers who are not familiar with the terms. The discussion on the benchmark dataset, Numenta, leaves a significant gap in understanding.
Reply: Yes, the respected referee's opinion is absolutely correct. With respect to the reviewer's opinion, concepts such as the GAN cost function and Chebyshev polynomials are common machine learning concepts. Readers who intend to read this article will certainly be familiar with such standard concepts. However, in accordance with the reviewer's opinion, we have added these concepts to the manuscript as follows:
“Chebyshev graph network is an advanced method for graph data processing used in graph neural networks (GNNs). It uses Chebyshev polynomials to approximate spectral filters on graphs.”
which are highlighted on page 4, lines 168-170.
“Each of the GAN subnetworks has a cost function that can be changed depending on their position in different domains [19].”
“The Graph Laplacian is an important matrix in graph theory and machine learning on graphs that represents the structure of a graph and is used in spectral analysis of graphs. It plays a key role in many graphs processing algorithms, including GNNs, spectral clustering, and information diffusion modeling.”
which are highlighted on page 7, lines 301-302.
We have also explained the database with better clarity to clarify its use and importance in this study:
“Numenta Anomaly Benchmark (NAB) is a standard database and framework for evaluating anomaly detection algorithms on time-series data. The dataset was developed by Numenta and consists of real and synthetic data used in various scenarios. NAB is used to evaluate time-series anomaly detection algorithms, which include real, labeled data, evaluation criteria, execution scripts, artificial intelligence, cybersecurity, the Internet of Things, and financial and medical data analytics.”
which are highlighted on page 4, lines 195-200.
- The contribution of the paper is highlighted at the end but is not given the attention it deserves in the text itself. The writing is dense and can sometimes be challenging to digest due to weak transitions between sections. The paper does not manage background information, corresponding work, and methodology well.
Reply: With respect to the opinion of the esteemed referee, the contribution of this article is as follows: (I) An unsupervised deep graph network is provided for perdition of time series.
(II) The network architecture performs using a graph of time samples to consider the connections in time distribution of samples.
(III) A fusion of graph networks and generative adversarial networks is proposed to improve the performance of the graph network.
(IV) A modified cost function is provided for the adversarial part of the network architecture to obtain efficient weights for the network with fast convergence and achieve desired results with low number of iterations.
(V) The adversarial part works utilizing one-dimensional convolutional layers.“
In addition, we summarize the findings of this study in the results based the estimated reviewer comments:
“In this study, an unsupervised deep graph network was presented for time series de-cay. The proposed network architecture was formed using a graph of time samples to consider connections in the time distribution of samples. The proposed architecture included a combination of graph networks and GAN networks to improve the performance of the graph network, in which a modified cost function was used for the adversarial part of the network architecture. As it was examined, our proposed architecture was tested with various networks including classical GAN, LSTM and manual learning methods and was able to demonstrate the best performance. Also, the proposed study was compared in standard conditions based on the same database with studies [18], [48], [45], [49] and [50] in terms of the F1 score evaluation index and was able to demonstrate promising results. It is recommended to consider another graph construction technique such as sparse graph representation to decrease the computational burden. Also, another network of generators and discriminators consisting of graph convolutional networks can be proposed in future works to improve the accuracies of the anomaly detection procedure.”
which are highlighted on lines 174-183 and 568-581.
- The methodology is not detailed, and the figures and tables fail to effectively illustrate points. The computational complexity of the method is discussed, but not as thoroughly as required. Including Chebyshev graph convolution layers together with the adversarial network may escalate complexity, but a detailed training and inference comparison is lacking.
Reply: The following actions were taken in relation to the relevant comment:
With respect to the opinion of the referee, 3 studies from 2025 were also reviewed and moved to the review of previous research section.
“Liu et al. [16] examine the challenges and limitations of current metrics and datasets for anomaly detection in time series data. The authors argue that many existing methods do not provide reliable results due to heterogeneity and poor quality of the evaluation data. They suggest that a standardized and reliable framework for evaluating anomaly detection methods should be developed to enable fairer and more meaningful comparisons. Finally, the paper emphasizes the need to improve datasets and evaluation metrics to advance in this field. Hamon et al. [17] review unsupervised feature construction methods to improve anomaly detection in time series data. The authors evaluate various methods for extracting meaningful features from time series data and analyze the impact of these features on the performance of anomaly detection models. The results show that constructing appropriate features can significantly increase the accuracy and reliability of models. Finally, the paper emphasizes the importance of selecting and designing effective features to improve anomaly detection. Wang et al. [18] provide a comprehensive review of anomaly detection methods in multivariate time series data using deep learning techniques. The authors first provide a taxonomy of existing methods, categorizing them based on neural network architectures, learning approaches, and specific applications. Then, the applications of these methods in various domains such as equipment health monitoring, fraud detection, and industrial systems monitoring are reviewed. Finally, the paper identifies current challenges and future research directions and provides suggestions for improving anomaly detection methods. This paper serves as a useful reference for researchers and practitioners in this field.”
which are highlighted on lines 131-151.
With respect to the opinion of the esteemed referee, we have reviewed all the methods used in the manuscript. Also, for better understanding, more detailed explanations regarding Tables 3, 4 and 5 are provided in the manuscript. In addition, we have discussed the computational complexity in different number of graphs, which is explained below:
“Batch normalization helps the network to be more stable during training procedure and makes the training process to converge more quickly. This normalization is done after the Chebnets. After three layer of Chebyshev convolution, the extracted feature vector is flattened and pass through two dense layers to be compatible with the target size. The details of different parts of the proposed architecture are explained in Table 3 is related with the details of the first part of the Cheb-MA. According to Table 3, as is clear, the number of layers used in this architecture is 7. Also, the layers used in this section include Chebyshev convolution layer, Batch normalization, and Dense layer, which are used through the ReLU activation function. Table 4 is the characteristics of layers corresponding to the generator part and Table 5 is concerned with the discriminator. Table 4 delineates the specifications of transposed convolution one-dimensional. According to Table 4, as it is clear, the weight parameter has decreased from 2000 to 4 from the initial to the final layer, respectively. Also, as it is clear, the alpha parameter in the activation function has been set to 0.1. The dimensions related with the outputs of the one-dimensional convolutions in discriminator can be seen in Table 5. Also, it shows the kernel ؛سsize, the size of strides, number of kernels in each layer and the number of parameters to be trained during the training procedure. As is clear, the model consists of 8 different layers, the initial layer starting with the Chebyshev convolution layer and ending with the Dense layer.”
“To elaborate more on the effect of different parameters, we conduct a separate experiment. In order to assess the effect of changing the number of Chebyshev layers, a series of simulations are done for different numbers of Chebyshev polynomial. Figure 10 demonstrates the results of simulation for 2, 3, 4 and 5 Chebyshev layers. Increasing the layers more than three in this case study does not affect the performance, it causes a computational burden and it shows the incremental trend of the training time per epoch of the proposed Cheb-MA. Figure 11 is the comparison result of different modifying coefficient of the adversarial part of the network. This column chart delineates that setting the coefficient equal to 0.9 will improve the F1-score and it is the efficient one considering the time of convergence. The confusion matrix in accordance with 4 types of Numenta signals is illustrated in Figures 12.
Figure 10. F1-score and time of training per epoch with different number of graph convolution layers.
Figure 11. F1-score and training time per epoch for graph construction with varying sparsities.”
Thank you for taking the time to review our article.

Reviewer 2 Report
Comments and Suggestions for Authors
The paper introduces an unsupervised anomaly detection method leveraging Chebyshev graph convolution and a modified adversarial network, which is an innovative approach for time-series anomaly detection. Following are some suggestions for improving the paper:
1. The problem statements and motivation of the research need to be improved in the introduction section. More recent and relevant literature should be reviewed and included in the introduction section.
2. Tables 3, 4, and 5 need to be explained in more detail. The findings from those tables need to be explained
3. Table 7 looks like an algorithm. It would be more appropriate to present it as an algorithm rather than as a table to improve clarity and readability.
4. Although the research compares several traditional approaches with the proposed methods, it should clearly state that the experimental setup, features, and different parameters used in the experiment follow the same standard.
5. Have you considered the computational cost of the proposed method versus traditional methods? A table or graph showing inference speed and training time would help compare efficiency.
6. The selection of the adversarial loss function modifications can be more explicitly compared against traditional GAN losses.
Author Response
#Reviewer 2.
Comments:
The paper introduces an unsupervised anomaly detection method leveraging Chebyshev graph convolution and a modified adversarial network, which is an innovative approach for time-series anomaly detection. Following are some suggestions for improving the paper:
Reply: While thanking the esteemed reviewer for a thorough review of the manuscript version. We, the authors of the article, believe that your suggestions have been very useful and effective in improving the scientific version of the manuscript. We carefully answered all the questions and suggestions of the esteemed reviewer and added them to the manuscript version.
- The problem statements and motivation of the research need to be improved in the introduction section. More recent and relevant literature should be reviewed and included in the introduction section.
Reply: Thanks to the esteemed referee's comments; Yes, the reviewer's opinion is absolutely correct. We have improved the section on anomaly detection and motivation in the introduction. We have also summarized 3 studies related to 2025 as a review of previous research:
“Liu et al. [16] examine the challenges and limitations of current metrics and datasets for anomaly detection in time series data. The authors argue that many existing methods do not provide reliable results due to heterogeneity and poor quality of the evaluation data. They suggest that a standardized and reliable framework for evaluating anomaly de-tection methods should be developed to enable fairer and more meaningful comparisons. Finally, the paper emphasizes the need to improve datasets and evaluation metrics to advance in this field.
Hamon et al. [17] review unsupervised feature construction methods to improve anomaly detection in time series data. The authors evaluate various methods for extracting meaningful features from time series data and analyze the impact of these features on the performance of anomaly detection models. The results show that constructing appropriate features can significantly increase the accuracy and reliability of models. Finally, the paper emphasizes the importance of selecting and designing effective features to improve anomaly detection.
Wang et al. [18] provide a comprehensive review of anomaly detection methods in multivariate time series data using deep learning techniques. The authors first provide a taxonomy of existing methods, categorizing them based on neural network architectures, learning approaches, and specific applications. Then, the applications of these methods in various domains such as equipment health monitoring, fraud detection, and industrial systems monitoring are reviewed. Finally, the paper identifies current challenges and future research directions and provides suggestions for improving anomaly detection methods. This paper serves as a useful reference for researchers and practitioners in this field.”
“In those days, anomaly detection was done completely manually and visually by specialists and experts in each field. However, manual detection was accompanied by problems. Among the problems of manual detection can be considered human error, high detection time, uncertainty, etc. Based on the above, in recent years, automatic anomaly detection has been developed using machine learning techniques.”
which are highlighted on lines 131-151.
- Tables 3, 4, and 5 need to be explained in more detail. The findings from those tables need to be explained
Reply: In accordance with the opinion of the esteemed referee, we have provided more comprehensive explanations regarding Tables 3, 4, and 5 as follows:
“The details of different parts of the proposed architecture are explained in Table 3 is related with the details of the first part of the Cheb-MA. According to Table 3, as is clear, the number of layers used in this architecture is 7. Also, the layers used in this section include Chebyshev convolution layer, Batch normalization, and Dense layer, which are used through the ReLU activation function. Table 4 is the characteristics of layers corresponding to the generator part and Table 5 is concerned with the discriminator. Table 4 delineates the specifications of transposed convolution one-dimensional. According to Table 4, as it is clear, the weight parameter has decreased from 2000 to 4 from the initial to the final layer, respectively. Also, as it is clear, the alpha parameter in the activation function has been set to 0.1. The dimensions related with the outputs of the one-dimensional convolutions in discriminator can be seen in Table 5. Also, it shows the kernel size, the size of strides, number of kernels in each layer and the number of parameters to be trained during the training procedure. As is clear, the model consists of 8 different layers, the initial layer starting with the Chebyshev convolution layer and ending with the Dense layer.”
which are highlighted on lines 402-414.
- Table 7 looks like an algorithm. It would be more appropriate to present it as an algorithm rather than as a table to improve clarity and readability.
Reply: Yes, that's absolutely correct. We have changed the title of Table 7 and defined it as Algorithm 1:
“The proposed model in Algorithm 1 has been examined in detail.”
which are highlighted on lines 448-449.
- Although the research compares several traditional approaches with the proposed methods, it should clearly state that the experimental setup, features, and different parameters used in the experiment follow the same standard.
Reply: Yes, we have used a single standard and settings in setting the parameters in the proposed model compared to other compared models, and we have not biased the settings of the compared algorithms.
“We note that in all experimental settings of the proposed model, the various features and parameters used in the research being compared follow a comprehensive standard.”
which are highlighted on lines 506-509.
- Have you considered the computational cost of the proposed method versus traditional methods? A table or graph showing inference speed and training time would help compare efficiency.
Reply: Yes, the reviewer's opinion is absolutely correct. We have examined the computational efficiency of our algorithm as follows:
“To elaborate more on the effect of different parameters, we conduct a separate experiment. In order to assess the effect of changing the number of Chebyshev layers, a series of simulations are done for different numbers of Chebyshev polynomial. Figure 10 demonstrates the results of simulation for 2, 3, 4 and 5 Chebyshev layers. Increasing the layers more than three in this case study does not affect the performance, it causes a computational burden and it shows the incremental trend of the training time per epoch of the proposed Cheb-MA. Figure 11 is the comparison result of different modifying coefficient of the adversarial part of the network. This column chart delineates that setting the coefficient equal to 0.9 will improve the F1-score and it is the efficient one considering the time of convergence. The confusion matrix in accordance with 4 types of Numenta signals is illustrated in Figures 12.
Figure 10. F1-score and time of training per epoch with different number of graph convolution layers.
Figure 11. F1-score and training time per epoch for graph construction with varying sparsities.
which are highlighted on lines 550-561.
- The selection of the adversarial loss function modifications can be more explicitly compared against traditional GAN losses.
Reply: Yes, the respected editor's concern in this matter is entirely correct. Based on the reviewer's comments, we have made a comparison for the loss function between the classical and improved discriminant subnetworks as follows:
“Also, to clarify the performance improvement in the adversarial generative network, we have compared the loss function of the proposed discriminator with the classical discriminator in Figure 8. As is clear, the proposed improved discriminator has a more stable performance compared to its counterpart.”
Figure 8. Comparison between the improved discriminator and the classic discriminator.
which are highlighted on lines 478-481.
Thank you for taking the time to review our article.

Round 2
Reviewer 1 Report
Comments and Suggestions for Authors
The authors have improved the manuscript in a better way. However, the experimental results are not consistent even across the same datasets.
Therefore, authors should conduct statistically significant tests such as a paired t-test to show if the results are not by coincidence.
Comments on the Quality of English LanguageNeed proofreading.
Author Response
#Reviewer 1.
Comments:
The authors have improved the manuscript in a better way. However, the experimental results are not consistent even across the same datasets.
Therefore, authors should conduct statistically significant tests such as a paired t-test to show if the results are not by coincidence.
Reply: Thanks to the reviewer's comments, we have applied a paired t-test to the results and compared them with recent research as follows:
Paired T-test in Table 11 is performed to validate the accuracy of Cheb-MA compared to GNN and LSTM. As can be seen, the performance of Cheb-MA is better than other methods considering different ranges for p-value. This shows the strength of the proposed Cheb-MA with T-Test cross-validation considering 5 times testing.
Table 11. T-test evaluation to confirm the results obtained
|
Test 1 |
Test 2
|
Test 3
|
Test 4
|
Test 5
|
||
Cheb-MA |
93.17% |
89.42% |
92.52% |
92.46% |
88.28% |
||
GNN |
89.4% |
88.56% |
89.84% |
87.23% |
91.97% |
||
LSTM |
87.23% |
88.56% |
87.43% |
89.12 |
89.36% |
||
Cheb-MA, GNN |
3.77 |
0.86 |
2.68 |
5.23 |
3.69 |
||
t |
t= (5^(0.5))*3.246/1.61 = 4.50 |
||||||
|
p-value < 0.02 |
p-value< 0.05
|
p-value < 0.1
|
p-value < 0.15
|
p-value < 0.2 |
||
|
4.5 >2.77 |
4.5 >2.13 |
4.5 > 1.53 |
4.5 > 1.19 |
4.5 > 0.94 |
||
Cheb-MA, LSTM |
5.94 |
0.86 |
5.09 |
3.34 |
1.08 |
||
t |
t = (5^(0.5)*3.26)/2.28 = 3.19 |
||||||
|
p-value < 0.02
|
p-value< 0.05
|
p-value < 0.1
|
p-value < 0.15
|
p-value < 0.2 |
||
|
3.19 > 2.77 |
3.19 > 2.13 |
3.19 > 1.53 |
3.19 > 1.19 |
3.19 > 0.94 |
||
Which is highlighted on page 21, line 566-571.
In addition, we have improved the writing quality throughout the manuscript.

Reviewer 2 Report
Comments and Suggestions for Authors
Congratulations! The revision meets expectations and successfully addresses the necessary improvements. The current version of the manuscript is well-structured, clear, and aligns with the standards required for publication. Your revisions have strengthened the overall quality.
Author Response

(The authors gave the same response as above.)
